# Stability of ZIF-8 Nanoparticles in Most Common Cell Culture Media

**DOI:** 10.3390/molecules27103240

**Published:** 2022-05-18

**Authors:** Anna S. Spitsyna, Artem S. Poryvaev, Natalya E. Sannikova, Anastasiya A. Yazikova, Igor A. Kirilyuk, Sergey A. Dobrynin, Olga A. Chinak, Matvey V. Fedin, Olesya A. Krumkacheva

**Affiliations:** 1International Tomography Center SB RAS, Novosibirsk 630090, Russia; a.spitsyna@alumni.nsu.ru (A.S.S.); poryvaev@tomo.nsc.ru (A.S.P.); sannikova.epr@gmail.com (N.E.S.); yaznast@mail.ru (A.A.Y.); 2N.Vorozhtsov Institute of Organic Chemistry SB RAS, Novosibirsk 630090, Russia; kirilyuk@nioch.nsc.ru (I.A.K.); dobrynin@nioch.nsc.ru (S.A.D.); 3Institute of Chemical Biology and Fundamental Medicine SB RAS, Novosibirsk 630090, Russia; chinakolga@gmail.com

**Keywords:** metal–organic frameworks (MOFs), ZIF-8, electron paramagnetic resonance (EPR)

## Abstract

Zeolite imidazolate framework-8 (ZIF-8) is a promising platform for drug delivery, and information regarding the stability of ZIF-8 nanoparticles in cell culture media is essential for proper interpretation of in vitro experimental results. In this work, we report a quantitative investigation of the ZIF-8 nanoparticle’s stability in most common cell culture media. To this purpose, ZIF-8 nanoparticles containing sterically shielded nitroxide probes with high resistance to reduction were synthesized and studied using electron paramagnetic resonance (EPR). The degradation of ZIF-8 in cell media was monitored by tracking the cargo leakage. It was shown that nanoparticles degrade at least partially in all studied media, although the degree of cargo leakage varies widely. We found a strong correlation between the amount of escaped cargo and total concentration of amino acids in the environment. We also established the role of individual amino acids in ZIF-8 degradation. Finally, 2-methylimidazole preliminary dissolved in cell culture media partially inhibits the degradation of ZIF-8 nanoparticles. The guidelines for choosing the proper cell culture medium for the in vitro study of ZIF-8 nanoparticles have been formulated.

## 1. Introduction

Zeolite imidazolate framework-8 (ZIF-8) belongs to a family of metal–organic frameworks (MOFs), formed by the coordination of Zn with 2-methylimidazole ligands (MIM) [1,2]. This porous material features a high surface area and adjustable pore size [3]. ZIF-8 can easily host a wide range of guests, including photosensitizers, nucleic acids, and proteins [4,5,6,7,8]. The unique properties of this MOF stimulated the development of flexible ZIF-8-based platforms for the drug delivery of small-molecule and macromolecular therapeutics [9]. ZIF-8 exhibits pH-controllable drug release, high efficiency of endosomal escape, and good biocompatibility [10,11,12,13,14,15,16,17,18]. The surface of ZIF-8 particles is modifiable by targeting agents, providing a higher efficiency of nanoparticle-based therapy [19,20,21].

ZIF-8 nanoparticles found application in cancer cells treatment, photodynamic therapy, and gene editing, as well as drug and protein delivery [10,22,23,24,25,26,27,28,29,30,31,32,33,34], for example, ZIF-8-based nanoparticles with polyacrylic acid and doxorubicin infiltrated breast cancer cells [35]. ZIF-8 demonstrated potential as a theranostic agent for cancer imaging [36]. ZIF-8 combined with cell-penetrating peptides enhanced the cellular uptake of oligonucleotides for gene therapy purposes [4]. ZIF-8 was also used for immunotherapeutics to deliver CpG oligodeoxynucleotides into immune cells [37]. Polyaniline-decorated ZIF-8 nanoparticles showed promising results for chemo-photothermal therapy of breast cancer cells [38]. Moreover, Xu et al. [39] constructed ZIF-8 containing water-insoluble photosensitizer zinc(II) phthalocyanine, in order to solve the aggregation problem for photodynamic therapy. The hydrophobic ZnPc molecules stored in the pores of ZIF-8 remained monomeric, avoiding self-aggregation in aqueous media.

The development of drug delivery systems, including the ZIF-8-based platforms, requires performing in vivo investigations, especially examining nanoparticle uptake by cells, as well as their cytotoxicity. Cell incubation with nanoparticles is performed in a cell culture media that contains a specially adapted set of nutrients and compounds. A medium provides survival and proliferation of cells but can affect experimental results [40,41]. Media with known compositions were developed to increase the reproducibility of studies, and many of them are commercially available.

It is known that ZIF-8 particles are rather unstable in serum and most common buffers [42,43,44]. At the same time, culture media contain various components that may also have significant effects on the stability of ZIF-8. For example, a greater degradation of ZIF-8 nanopowders in bacterial culture media than in deionized water was recently demonstrated [45]. The most popular cell medium used in experiments with ZIF-8 is DMEM [35,37,46,47]. It was previously shown that DMEM leads to at least a partial degradation of ZIF-8 microparticles [42] and nanoparticles [48]. Besides DMEM, other media, such as RPMI-1640, L-15, and IMDM, have also been applied in cell experiments with ZIF-8 [36,38,49]. Given the significant differences in the composition of these media, the extent of the degradation of ZIF-8 nanoparticles may also vary.

To the best of our knowledge, the systematic investigation of ZIF-8 stability in various cell culture media has not been performed, despite the fact that this information is essential for in vitro studies. To obtain generalized and quantitatively useful insight into these issues, in this work, we systematically investigate the stability of ZIF-8 nanoparticles in a set of commonly used cell culture media.

Recently we proposed a new approach for quantitatively probing the extent and rate of guest release from ZIF-8 particles directly under hands-on conditions by EPR [43]. Tracking the cargo leakage from ZIF-8 upon dissolution of the framework was carried out using a nitroxide spin probe. The key point of the method is that the EPR spectra of nitroxide inside and outside ZIF-8 are noticeably different. The shape of the EPR spectrum of the studied sample represents the distribution of the probe between ZIF-8 particles and the solution. The simulation of the experimental data provides the quantification of the cargo leakage. Generally, this approach is easily applicable to studying ZIF-8 in any solution, including biological media. However, it should be taken into account that the tetramethyl-substituted nitroxide used in our previous work is rather unstable in some biological systems. For instance, the radical moiety of a nitroxide probe is subject to reduction by some components of cell media (e.g., ascorbic acid and glutathione), yielding diamagnetic hydroxylamines that are invisible by EPR. To overcome this problem, in this work, we used an advanced, sterically shielded spin probe based on tetraethyl-substituted nitroxide R (Figure 1a) with a high resistance to reduction [50].

## 2. Results and Discussion

### 2.1. Synthesis and Characterization of ZIF-8 Nanoparticles

To study the ZIF-8 stability in cell media, we synthesized ZIF-8 nanoparticles with an encapsulated spin probe R (3,4-Bis-(hydroxymethyl)-2,2,5,5-tetraethylpyrrolidin-1-oxyl), according to the previously-described procedure [51]. Dynamic light scattering (DLS) measurement showed that the mean size of R@ZIF-8 nanoparticles is 180 nm (Appendix A), which is the typical size of nanoparticles in biomedical applications [52].

According to continuous wave (CW) EPR data, R@ZIF-8 nanoparticles contain 0.4 nitroxides radical per 1 ZIF-8 cavity, mimicking highly-loaded drug carriers. The initial suspension of R@ZIF-8, with about 6.5 mg/mL concentration, demonstrates a significantly broadened spectrum with three overlapping lines (Figure 2, top trace). We have previously shown that the EPR line broadening for probes inside highly-loaded ZIF-8 is caused by the exchange and dipolar interactions between the radicals located in the same or neighboring cavities of ZIF-8 [43].

Figure 2 compares the EPR spectra of the initial suspension of R@ZIF-8 nanoparticles (top trace, 6.5 mg/mL) and its five-fold diluted solutions (all other traces), with a final concentration 1.3 mg/mL. Besides pure water, we also examined R@ZIF-8 nanoparticles in standard buffers, such as phosphate buffered saline (PBS, 5 mM, pH = 7.4), 4-(2-hydroxyethyl)-1-piperazineethanesulfonic acid (HEPES, 5 mM, pH = 7.4), and citrate buffer (0.1 M pH = 4.0). All spectra of the diluted samples in Figure 2 have the same second integral, confirming that the total concentration of spins is conserved; however, the shape of the spectra are significantly diverse from each other. The EPR spectrum of the R@ZIF-8 mixed with acid buffer at pH = 4.0 completely differs from the initial one: it consists of three narrow lines and coincides with the spectrum of a free R in water (Appendix A). In this case, the full release of cargo results from the complete decomposition of the ZIF-8 framework in acid environments.

All experimental spectra presented further were simulated as a superposition of the normalized spectra of R@ZIF-8 in initial suspension and acid pH. The weights of these two contributions represent the amount of probes that escaped from dissolved ZIF-8, as well as those that remained located inside initial nanoparticles.

The dilution of initial R@ZIF-8 suspension by pure water induced a minor (~1%) release of guest molecules, and the spectrum remained the same during further storage over 1–2 days (Figure 2, Appendix A). The EPR spectra of R@ZIF-8 mixed with 5 mM of PBS or HEPES have another shape and feature, i.e., three notable narrow lines superimposed on the broad component. According to the simulation, about 10% and 25% of cargo escaped from ZIF-8 in HEPES and PBS, respectively. These results confirm the previously-described degradation of ZIF-8 particles in the presence of buffer components [42,44]. We have previously shown that the release of cargo from ZIF-8 in buffer-containing solutions is a concentration-dependent process [43]. Therefore, the concentrations of buffer components should be carefully considered when studying the ZIF-8 stability in culture media.

### 2.2. Dissolution of ZIF-8 in Cell Culture Media

There are many different types of cell culture media, but we have chosen those most commonly used in modern research for our study. One of the basic media is Eagle’s minimum essential medium—MEM. Several other compositions, such as DMEM, α-MEM, and IMDM, were developed based on MEM. Opti-MEM is another improved MEM formula that is recommended for use with cationic lipid transfection reagents. Its complete composition is confidential. Besides the MEM media family, we also studied the following media: RPMI-1640, which contains the reducing agent glutathione; L-15, which contains no sodium bicarbonate and, therefore, does not require a CO_2_ environment to maintain physiological pH; and 199, supplemented with nucleobases ATP and AMP. The full composition of studied media is listed in Appendix A.

The decomposition of R@ZIF-8 upon five-fold dilution by different media was monitored by CW EPR. An immediate (<1 min) leakage of a portion of guest molecules was observed after the media addition, which was not followed by further particle degradation. All detected spectra had three noticeable narrow lines, indicating that all cell media caused considerable decomposition of ZIF-8 nanoparticles (Appendix A). A quantitative analysis of the contributions of the broad and narrow components showed that the degradation effect depends on the type of medium (Appendix A). The moderate destabilization effects were observed in 199, MEM, and α-MEM, where the fraction of released probes was about 25–50% (green area in Figure 3). In contrast, we observed an almost complete dissolution of R@ZIF-8 in RPMI-1640, IMDM, and L-15 (red area in Figure 3). The most frequently used DMEM media and opti-MEM featured an intermediate result; in this case, about 70% of probes leaked out of the ZIF-8 framework (yellow area in Figure 3).

The extent of the ZIF-8 degradation rose upon decreasing the mass ratio of nanoparticles to cell media. We observed the full release of the probe already, with a mass ratio of R@ZIF-8 equal to 0.65 mg/mL in MEM, exhibiting one of the best nanoparticle stability results at higher concentrations (Appendix A).

All studied media, except RPMI-1640, contained a low concentration of phosphates (1 mM or less) and similar amounts of sodium bicarbonate (2.6–4.4 mM, Appendix A). This means that the dissolution of ZIF-8 nanoparticles by buffer components cannot be the main pathway for nanoparticles degradation in these media. Hence, the observed diversity of the extent of released cargo upon dissolution of ZIF-8 in different media should be associated with other factors.

In addition to the buffer components, cell media also contain high concentrations of various amino acids (Appendix A). Therefore, we analyzed how the total concentration of amino acids in the media correlates to the extent of ZIF-8 degradation. Figure 3 illustrates the general trend toward a decreasing R@ZIF-8 amount with increased amino acid concentrations. In particular, L-15 was the most amino acid-rich media in our set, and it exhibited the most remarkable ability to dissolve R@ZIF-8. At the same time, the relatively high stability is realized in the 199 and MEM, where the concentration of amino acids is about four times less than in L-15. Thus, amino acids make a considerable contribution to the decomposition of ZIF-8 nanoparticles.

RPMI-1640 relates to the special case, since its ability to induce cargo leakage from ZIF-8 nanoparticles is significantly higher than for α-MEM, despite the similar amino acid content in both media (Figure 3). This means that additional contributions may also affect the stability of the nanoparticles. Unlike other cell media, RPMI-1640 contains a high concentration of PBS components (5.6 mM) that, as shown above, induce the leakage of probes from ZIF-8 nanoparticles. The dissolution by amino acids and buffer components, both being effective degradation pathways of ZIF-8, leads to an almost complete release of cargo in RPMI-1640. In addition, another media, IMDM, contains 25 mM of HEPES buffer, which also affects the stability of ZIF-8 nanoparticles, although to a lesser extent than PBS. Thus, there are at least two contributions to ZIF-8 destabilization in these media: dissolution by amino acids and buffer components.

In addition, most culture media require supplementation with 10% fetal bovine serum (FBS). Previously, it was shown that ZIF-8 undergoes decomposition in the FBS [43]. However, in this work, we observed only a minor release of cargo (~10%) in the presence of 10% FBS. This indicates that the contribution of FBS to the dissolution of ZIF-8 nanoparticles is negligible, compared to other pathways, at a given concentration.

### 2.3. Dissolution of ZIF-8 by Individual Amino Acids

The studied media differ not only in the total concentration of amino acids, but their ratios also vary significantly in the final composition. To deeply understand the role of different amino acids in ZIF-8 dissolution, we investigated EPR spectra of R@ZIF-8 five-fold diluted in HEPES (5 mM, pH = 7.4) containing 5 mM of individual amino acids (Appendix A). Figure 4a shows the results of studying the nanoparticle degradation caused by 20 compounds, covering all classes of amino acids.

The obtained quantitative data must be considered relative to the effect in pure HEPES, where the fraction of probes leakage was ~10% upon dissolution of R@ZIF-8. All amino acids affect R@ZIF-8 stability, but the fraction of released cargo differs significantly (Appendix A). The dissolution of R@ZIF-8 in the presence of 5 mM of proline, lysine, arginine, and alanine induced only a slight cargo leakage (≤10%), relative to the value in HEPES (blue area in Figure 4a). The opposite effect was observed for cysteine, where almost all probes escaped from ZIF-8 at the same concentrations (red area in Figure 4a). In addition, we observed a significant degradation of nanoparticles in the presence of histidine and phenylalanine, where only 40–50% of the cargo remained inside the ZIF-8 upon dissolution (yellow area in Figure 4). Other amino acids exhibited intermediate results: they noticeably dissolved nanoparticles, but most of the probes (>60%) did not leave the cavities (green area in Figure 4a).

We also measured the concentration dependence of the released probes for several amino acids, namely two compounds with high (cysteine and histidine) and moderate (lysine and serine) destabilization effects on ZIF-8 (Appendix A). Figure 4b shows the relation between the amount of cargo held in ZIF-8 after dissolution and amino acid concentrations. The number of released probes rises upon increasing amino acid concentrations in all cases (Appendix A). However, it occurs considerably faster for cysteine and histidine than for lysine and serine.

It is known that Zn^2+^ and MIM ligands are released from ZIF-8 in water [53,54]. In addition, our results showed that the dissolution of ZIF-8 by amino acids is a concentration-dependent process (Figure 4b). The maximum amount of zinc ions that could be released upon complete dissolution of ZIF-8 was around 5 μmol/mL in our experiments, whereas the total concentration of amino acids and phosphates in cell media was at least the same (for MEM, 199, and alpha-MEM, Appendix A) or even higher (for RPMI, DMEM, IMDM, and L15, Appendix A). Therefore, we assumed that the degradation of nanoparticles is caused by the binding of zinc ions with the amino acids. To verify this, we analyzed the correlation between the amount of released probes in different amino acid solutions and published stability constants for the complexes of these amino acids with zinc ions [55]. The stability constants (β) describe the following model reaction: Zn^2+^ + 2AA ↔ Zn(AA)_2_, where AA represents an amino acid in deprotonated form. We also considered that the amino acids have different dissociation degrees at the used pH = 7.4, depending on the acidity constant K_a_. Hence, the zinc complexation in amino acid solutions depends on two constants: β and K_a_, listed in Appendix A. Thus, we examined the relationship between ZIF-8 degradation and log(βK_a_^2^) in amino acid solutions. Figure 5 reveals strong positive correlations between the amount of released probe and log(βK_a_^2^). The high degree of ZIF-8 degradation in histidine and cysteine correlates well with the higher stability constant for complexes of these amino acids with zinc ions, rather than for the other studied amino acids (more than 100 times) [55].

High values of histidine and cysteine complexation with zinc ions can be explained by the chemical nature of amino acids in the following way. In case of histidine, zinc ions forms a six-membered chelate ring through coordination with two nitrogen atoms of the histidine moiety that could be more stable than analogous one with other amino acids [56]. In contrast, other amino acids form a five-membered chelate ring, through coordination with nitrogen and oxygen [56]; therefore, their complexes with zinc ions are less stable compared to histidine. The sulphur atoms of cysteine have high affinity to zinc ions [57,58]. Thus, cysteine also exhibits the high stability of complexes with zinc ions, despite the five-membered chelate ring.

### 2.4. Stabilization of ZIF-8 Nanoparticles by 2-Methylimidazole

The aforementioned mechanism of particle degradation due to the binding of amino acids with zinc is supposed to release Zn^2+^ and 2-methylimidazole ligands (MIM) from ZIF-8 nanoparticles at the first stage. As a result, ZIF-8, Zn^2+^, and MIM reach an equilibrium state in the solution. Therefore, we assume that the preliminary addition of MIM ligands into cell media can suppress the dissolution of ZIF-8 nanoparticles, due to a shift in equilibrium towards zinc-binding with MIM, rather than with amino acid.

To validate this hypothesis, we first investigated MIM’s effect on the stability of R@ZIF-8 upon dilution, with the amino acids exhibiting a moderate or strong ZIF-8 degradation effect. The stabilization effect was explored in the HEPES buffer (5 mM, pH 7.4) containing 5 mM of amino acid, 10 mM of MIM, and 1.3 mg/mL of R@ZIF-8.

We observed the considerable stabilization of ZIF-8 nanoparticles in the presence of MIM for almost all examined amino acids (Figure 4a and Appendix A). For example, the addition of ligand led to a decrease of released cargo, from 40% to 20% in the asparagine solution. The inhibitive effect of MIM confirms the hypothesis that the decomposition of ZIF-8 nanoparticles in cell media can be described as an equilibrium between the ZIF-8 and Zn^2+^ complexes with other ligands. In contrast, MIM did not demonstrate any stabilization effect in the histidine and cysteine solutions, probably because of the strong complexation of zinc ions in their presence.

Next, we performed similar experiments in the cell media. A total of 10 mM of MIM inhibited ZIF-8 degradation in DMEM, opti-MEM, and L-15, decreasing the released cargo by 10% or more (Figure 6a,b and Appendix A). A further increase of the concentration of MIM, up to 30 mM, leads to an additional drop of the leaked probes by about an extra 10%. Unlike other cell media, DMEM and L-15 contain a large concentration of amino acids and, at the same time, moderate (0–0.9 mM) concentration of PBS. Hence, the added MIM effectively competes with amino acids for binding with Zn^2+^, thus stabilizing the nanoparticles.

The effect of MIM on ZIF-8 stability is smaller for RPMI-1640 and IMDM (Figure 6a). These media have high amounts of buffer components—5.6 mM of PBS in RPMI-1640 and 25 mM of HEPES in IMDM. Therefore, the contribution of buffer components to the ZIF-8 dissolution is higher than in other media. Our results indicate that MIM can inhibit the dissolution of ZIF-8 by amino acids but, to a lesser extent, also affect the degradation pathways related to the presence of a buffer. This is also supported by data for 199, MEM, and α-MEM. As shown above, amino acid’s contribution to the degradation of nanoparticles is the smallest in 199, MEM, and α-MEM (among the whole set of cell media). As a result, the stabilization by MIM only slightly manifests itself in these cases.

Additionally, we have shown that 24 h incubation of cells with 10 mM MIM does not decrease cell viability, and there is a moderate cytotoxic effect at 30 mM (60–70% cell viability for 24 h cell incubation, Appendix A).

## 3. Conclusions

We have performed a quantitative investigation of the ZIF-8 nanoparticle’s stability in the most common cell culture media. Our data indicate that the ZIF-8 nanoparticles at least partially degrade in all of the studied culture media, although the degree of cargo leakage varies widely, depending on composition of cell media. We found a strong correlation between the amount of escaped cargo and total concentration of amino acids in the environment. Thus, using cell media with the lowest amino acid content, such as 199, MEM, and α-MEM, is preferred in studies of ZIF-8 nanoparticles. In comparison, the L-15 tends to dissolve ZIF-8 nanoparticles to the greatest extent, because of the high concentration of amino acids. Media with a high PBS or HEPES concentration, namely RPMI-1640 and IMDM, should also be avoided. The cumulative effect of dissolution by amino acids and buffer components in these cell media causes severe destabilization of ZIF-8 nanoparticles and considerable cargo release.

We also established the role of individual amino acids in ZIF-8 dissolution. We found that the amount of cargo escaped from ZIF-8 depends on the stability constants for the amino acids with zinc ions. The strongest dissolution effects were observed in the histidine and cysteine solutions, which was determined by the high stability constants of their complexes with zinc ions. Therefore, the concentrations of these two amino acids in cell media should be carefully considered in the experiments with ZIF-8 nanoparticles in vitro.

We also found that MIM partially inhibits ZIF-8 dissolution in the DMEM, opti-MEM, and L15 at 10 mM concentration, and it is nontoxic for cells at this amount. Moreover, 30 mM of MIM showed a moderate cytotoxicity for 24 h cell incubation. Thus, MIM can be safely used for ZIF-8 stabilization at these concentrations in short-term cellular studies. Finally, the preliminary addition of 30 mM of MIM to the most widely used DMEM medium significantly reduces cargo release, to the same level as in MEM, exhibiting one of the best levels of nanoparticle stability.

## 4. Experimental

### 4.1. Materials

All solvents used were of HPLC-grade quality. All reagents and solvents were purchased from commercial sources and used as received, without further purification. Buffers were purchased from Sigma-Aldrich (Burlington, MA, USA).

MEM alpha powder (with ribo- and deoxyribonucleosides, IMDM powder (with L-glutamine); DMEM (with L-glutamine, glucose, pyridoxine, and pyruvate) by Gibco (Burlington, MA, USA); MEM (with L-glutamine), RPMI-1640, and L-15 powder (with L-glutamine) by Sigma-aldrich; opti-MEM (with L-glutaMAX) by Gibco (USA); medium 199 by Capricorn (Ebsdorfergrund, Germany). All powder media were diluted according to protocol and filtered (0.22 µm).

All L-amino acids and MTT (3-(4,5-dimethyl-2-thiazolyl)-2,5-diphenyl -2H-tetrazolium bromide by Sigma (USA).

### 4.2. Synthesis of R

The sterically shielded nitroxide was prepared according to the literature procedure [50].

### 4.3. Synthesis of R@ZIF-8

R@ZIF-8 nanoparticles were prepared according to the procedure described previously [20,51]. For the synthesis of ZIF-8, nanoparticle R was dissolved in 2.5 M MIM solution to concentration around 0.0027 M, and the mixture was stirred for 30 min. Then, 0.5 M zinc nitrate solution was slowly added to the above mixture under mechanical agitation for 30 min. ZIF-8 nanoparticles were collected by centrifugation and washed with water 3 times. PXRD measurements were carried for the radical@ZIF-8 samples in previous studies [20,51], and phase purity of radical@ZIF-8 samples was approved.

Freshly prepared ZIF-8 suspension has been stored at room temperature for one week; during this time, EPR measurements were carried out. The quality of the ZIF-8 nanoparticles in the initial suspension was monitored every day by EPR, and we did not observe any degradation for one week. All presented results were reproduced several times using different freshly prepared ZIF-8 suspensions.

### 4.4. Sample Preparation

The amino acids were dissolved in the HEPES buffer (5 mM, pH = 7.4) with 5 mM concentration, except Asp (2.5 mM), Tyr (1 mM). In order to study the concentration dependence of R@ZIF stability the Lys, Ser, His, and Cys were dissolved in the same buffer, with concentrations 1, 2.5, 5, and 10 mM.

To study stabilization by MIM, we added a 5% (15%) of 0.2 M solution of MIM to the several media. The final MIM concentration was 10 mM (30 mM). Then, 20% (by volume) of the nanoparticle suspension was added to the final solutions.

The pH of all solutions was controlled and titrated to 7.4 with NaOH or HCl at every sample preparation step. Measurement of pH was performed using a pH electrode, InLab Micro (Mettler Toledo, Greifensee, Switzerland), calibrated with buffer standards (pH 4, 7, and 10). We did not consider a possible impact of Zn^2+^ complexation by chloride in this work, because the binding constant β of such a complex is small (Log(β3) < 2.5) [59].

### 4.5. Characterization of ZIF-8 Particles

The size distribution of ZIF-8 particles was analyzed using a Zetasizer Nano, series Nano-ZS (Malvern Instruments, Malvern, UK).

### 4.6. EPR Measurements

Continuous wave (CW) EPR spectra were obtained at X-band and room temperature via a commercial X-band Bruker (Billerica, MA, USA) EMX spectrometer. Samples were placed in glass capillary tubes (OD 1.5 mm, ID 0.9 mm). Spin concentration was determined by comparing the spectral double integrals of the sample and a nitroxide solution with a known concentration. CW EPR spectra were recorded at conditions that avoided unwanted modulation broadening and microwave saturation. Experimental CW EPR settings were as follows: sweep width, 20 mT; microwave power, 6.315 μW; modulation frequency, 100 kHz; modulation amplitude, 0.4 mT; time constant, 20.48 ms; conversion time, 81.92 ms; the number of points, 1024; number of scans, 4. The baseline spectrum of HEPES buffer was acquired with 80 scans. The baseline spectrum was subtracted from all of the spectra.

Spectra were simulated using a home-written MATLAB script with the EasySpin toolbox [60], as a superposition of the normalized spectra of R@ZIF-8 in initial suspension (spectrum component S1) and acid pH = 4.0 (spectrum component S2): *S* = *αS*1 + *βS*2. The corresponding weights (*α* and *β*) were determined using the least squares regression of experimental and simulated data. Linear function, as a baseline, was subtracted from the first integral of spectra.

### 4.7. Cytotoxic Assay of MIM

A549 cells were seeded in 96-well plates at a density of 2 × 10^3^ cells per well by 100 µL DMEM supplemented with 10% FBS, antibiotics (100 U/mL penicillin, 100 mg/mL streptomycin, 0.25 μg/mL amphotericin B) and 2 mM l-glutamine. After 24 h, 100 µL of DMEM containing antibiotics, 2 mM l-glutamine, and MIM per well were added. After 24 h, the media were replaced with 200 mL of RPMI-1640 medium containing 0.25 mg/mL MTT(3-(4,5-dimethyl-2-thiazolyl)-2,5-diphenyl-2Htetrazolium bromide), and the cells were incubated at 37 °C for 4 h. Cell viability was expressed as a mean percentage of control ± SD for triplicate independent experiments.

## Figures and Tables

**Figure 1 molecules-27-03240-f001:**
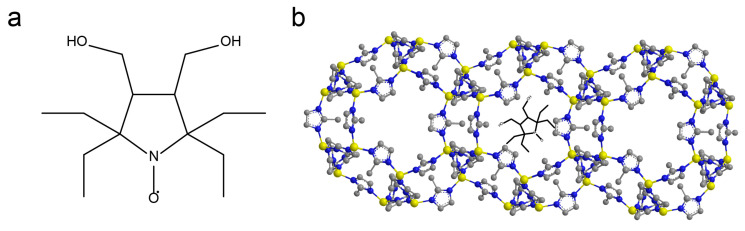
(**a**) Structure of stable nitroxide radical R used as a guest molecule; (**b**) ZIF-8 structure with incorporated radical R.

**Figure 2 molecules-27-03240-f002:**
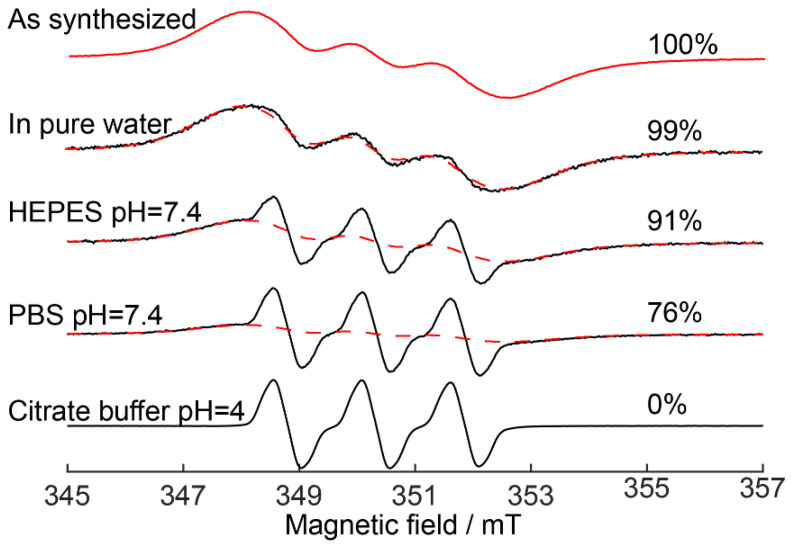
EPR spectra of initial suspension of R@ZIF-8 (as synthesized), R@ZIF-8 5-fold diluted in PBS (5 mM, pH = 7.4), HEPES (5 mM, pH = 7.4), and citrate buffer (0.1 M, pH = 4.0). Red dashed lines show the broad component in the spectra that corresponds to the probes located inside ZIF-8. The number on the right shows the amount of probe remaining inside ZIF-8 after dissolution.

**Figure 3 molecules-27-03240-f003:**
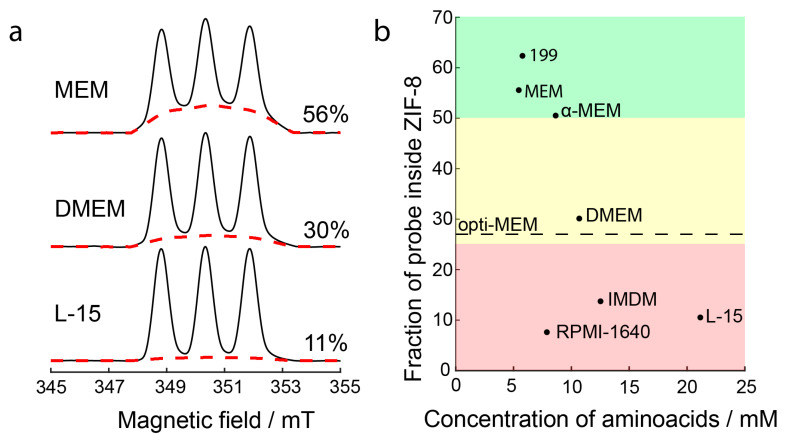
Dissolution of ZIF-8 in cell culture media. (**a**) First integrals of CW EPR spectra of R@ZIF-8 in culture media. Red dashed lines show the spectra obtained by simulation for a fraction of the probe inside ZIF-8. The numbers on the right demonstrate the weight of this fraction; (**b**) the fraction of probes inside ZIF-8 versus total amino acids concentration in studied media. Stability is denoted by color-coding: red—most nanoparticles dissolved, yellow—middle situation, and green—half or more nanoparticles intact. The result for opti-MEM is demonstrated as a dotted line, because its complete composition and concentration of amino acids are confidential.

**Figure 4 molecules-27-03240-f004:**
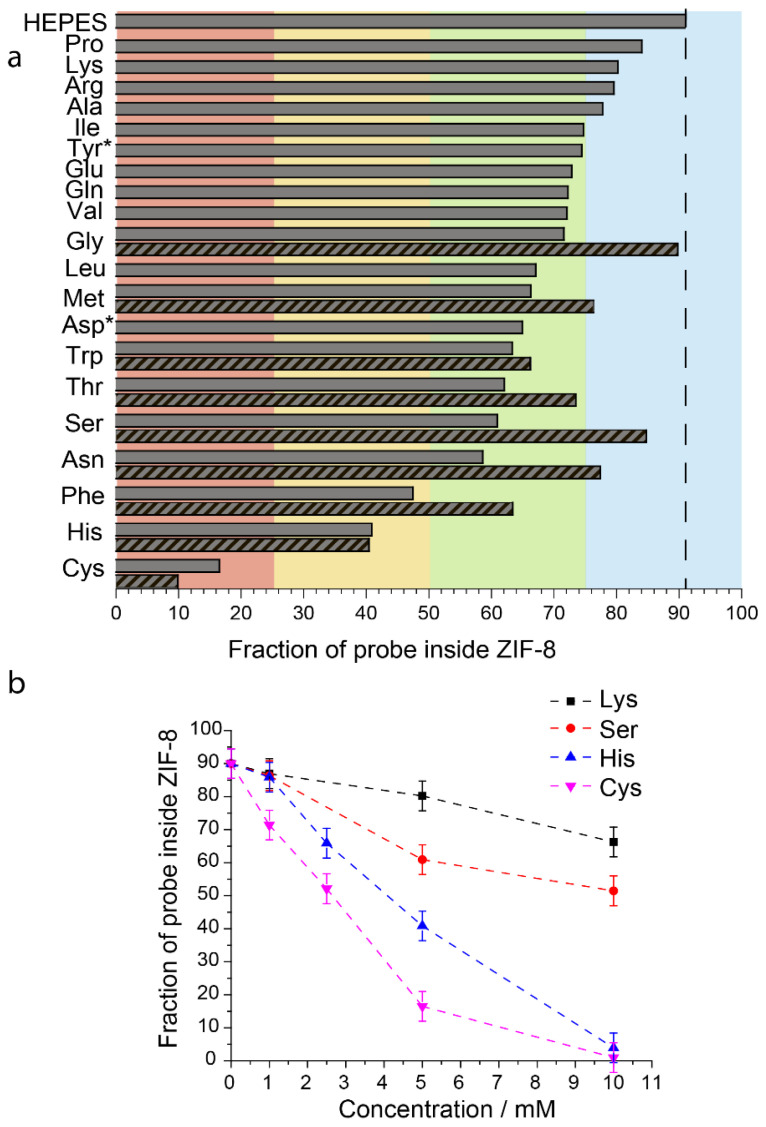
(**a**) Fraction of the probe inside ZIF-8 in R@ZIF-8 sample 5-fold diluted in HEPES (5 mM, pH = 7.4) containing 5 mM of individual amino acids. Hatching demonstrates samples with 10 mM MIM preliminary added to media. * marks two amino acids with different concentrations—the highest we achieved considering the solubility of these amino acids in water (Appendix A); (**b**)—Dependence of the probe inside ZIF-8 on amino acids concentration added.

**Figure 5 molecules-27-03240-f005:**
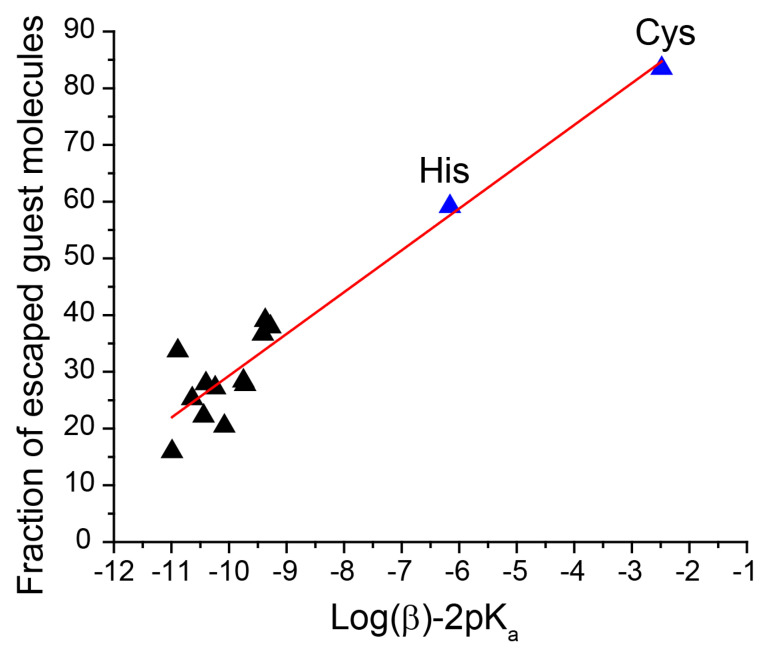
Correlation between a fraction of spin probes escaped from ZIF-8 and factor Log(β)-2pK_a_, being a combination of stability constant with zinc and pK_a_ for amino acids (Appendix A).

**Figure 6 molecules-27-03240-f006:**
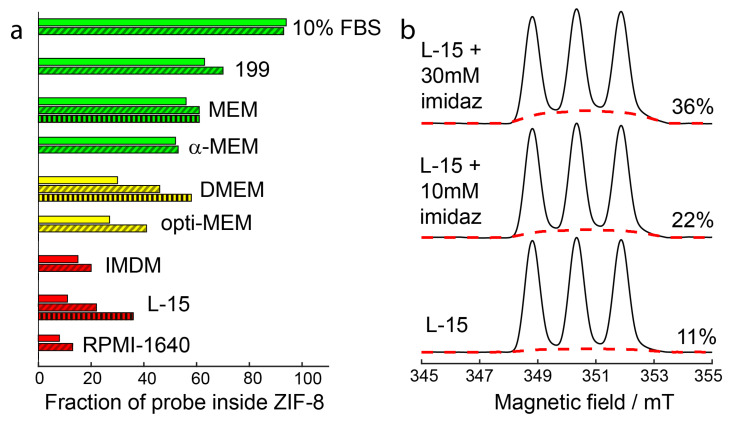
(**a**) Changes in fraction of probe inside ZIF-8, upon addition of different cell media (Appendix A). Oblique hatching marks samples with addition of 10 mM MIM, vertical hatching—addition of 30 mM MIM; (**b**) first integrals of CW EPR spectra for R@ZIF-8 with addition of L-15 and MEM. Red dashed lines show the fraction of radicals remaining in ZIF-8; numbers on the right demonstrate the weight of this fraction.

## Data Availability

Not available.

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
