# Peer review of "Stability of ZIF-8 Nanoparticles in Most Common Cell Culture Media"

_molecules, 2022, doi:10.3390/molecules27103240_

Round 1

Reviewer 1 Report

The paper describe the potential of Zeolite imidazolate framework-8 (ZIF-8) as a promising platform for drug delivery and therefore conduct a quantitative investigation of ZIF-8 nanoparticle's stability in most common cell culture media. With this purpose, ZIF-8 nanoparticles containing sterically shielded nitroxide probes with high resistance to reduction were synthesized and studied using Electron Paramagnetic Resonance (EPR) spectroscopy.

I found this research interesting and very well performed. I recommend to publish this paper.

Author Response

We thank the reviewer for the positive feedback.

Reviewer 2 Report

The authors reported a quantitative investigation of the ZIF-8 nanoparticle's stability in various cell culture media. The ZIF-8 nanoparticles partially degrade in all studied culture media, and the degree of cargo leakage varies widely depending on composition of cell media. The amino acids influence the stability greatly and causes severe destabilization of ZIF-8 nanoparticles due to the complex of Zn2+ with amino acids. The author also found that 2-Methylimidazole could stabilize the ZIF-8 based on the consideration of equilibrium state in the solution. The research is interesting and is helpful for the in vitro study of nanomedicine with ZIF-8 or other materials. The English is well written and the paper is well organized.

I recommend its publication in Molecules with after minor revision

  1. Writing errors. Page 9 line 327

2. The reason for investigating the influence of amino acids should be given. At least some background description should be added in the paragraph in page 6 line 199. 

Author Response

Point 1. Writing errors. Page 9 line 327

Reply:

We have corrected these errors in the text, thank you

The new text reads:

“All L-amino acids and MTT (3-(4,5-dimethyl-2-thiazolyl)-2,5-diphenyl -2H-tetrazolium bromide by Sigma (USA).”

Point 2.. The reason for investigating the influence of amino acids should be given. At least some background description should be added in the paragraph in page 6 line 199. 

Reply:

We have re-written the in the paragraph in page 6 line 199 and add the description of the reason for investigating the influence of individual amino acids.

The new text reads:

“The studied media differ not only in the total concentration of amino acids but also their ratios vary significantly in the final composition. To deeply understand the role of different amino acids in ZIF-8 dissolution, we investigated EPR spectra of R@ZIF-8 5-fold diluted in HEPES (5 mM, pH=7.4) containing 5 mM of individual amino acids (Figure S6).”

Reviewer 3 Report

The paper by Spitsyna et al. reports the investigation of ZIF-8 stability in several vital-related media. The work possesses high novelty, it is carried out on a well scientific level and obviously deserves interest of the Molecules wide readership. However, some experimental data and discussions should be presented more accurately. Below are some critical notes to be considered by the authors: 

1. MOFs are usually metastable kinetic products, but not thermodynamically stable. In this regard, I am not sure if it is correct to describe the decomposition process as an equilibrium between ZIF-8 and Zn complexes with other ligands (amino acids etc). Despite the conclusions obtained from Keq estimations are beyond doubt, authors should describe in the text how long the prepared ZIF-8 suspensions have been stored before EPR measurements and is all this preparation procedure reproducible. The concentration units need to be added into table S1 and an approximate stoichiometry between ZIF-8 and buffer constituents should be considered - is the buffer in a real excess over MOF? Also, a possibile impact of Zn(II) complexation by chloride (lines 347-348) should be commented. 

2. I guess authors could be a little more generous in the discussion of amino acid-driven decomposition of ZIF-8. May the highest extent of ZIF-8 destroy by histidine and cysteine be explained in terms of their chemical nature?

3. A purity and composition of the obtained R@ZIF-8 phase needs to be proven by some conventional characterization, e.g. PXRD/IR/CHN data. 

4. Some references should be provided to confirm that 60-70% cell viability represents "only a slight cytotoxicity" (line 288). 

Some minor corrections: 

5. Author list does not match in the manuscript and ESI. 

6. Line 152: dilution → dissolution

7. Line 166: The introductory phrase is probably missing

8. pH-meter should be described in the experimental

Author Response

We thank the reviewer for the positive feedback.

Point 1. MOFs are usually metastable kinetic products, but not thermodynamically stable. In this regard, I am not sure if it is correct to describe the decomposition process as an equilibrium between ZIF-8 and Zn complexes with other ligands (amino acids etc).

Reply:

We have studied kinetics of guest release from ZIF-8 suspension previously. We have shown that guest release even for microparticles occurs immediately[10.1021/acs.jpcc.1c03876]. Thus, we can use quasi-equilibrium approximation. In addition, the inhibitive effect of MIM also confirms the hypothesis that the decomposition of ZIF-8 nanoparticles in cell media can be described as an equilibrium between ZIF-8 and Zn complexes with other ligands. Consequently, we think that performed analysis of experimental data is reasonable. To clarify this point we added additional explanation into the text.

The new text reads:

“The inhibitive effect of MIM confirms the hypothesis that the decomposition of ZIF-8 nanoparticles in cell media can be described as an equilibrium between ZIF-8 and Zn complexes with other ligands.”

Point 2. Despite the conclusions obtained from Keq estimations are beyond doubt, authors should describe in the text how long the prepared ZIF-8 suspensions have been stored before EPR measurements and is all this preparation procedure reproducible.

Reply:

The required preparation procedures have been added into Experimental part 

The new text reads:

“Freshly prepared ZIF-8 suspension has been stored at room temperature for one week and during this time EPR measurements were carried out. The quality of the ZIF-8 nanoparticles in the initial suspension was monitored every day by EPR and we did not observe any degradation for one week. All presented results were reproduced several times using different freshly prepared ZIF-8 suspensions.”

Point 3. The concentration units need to be added into table S1 and an approximate stoichiometry between ZIF-8 and buffer constituents should be considered - is the buffer in a real excess over MOF?

Reply:

We added the concentration units in Table S1 and information about buffer constituents in the main text. The total amount of zinc ions in ZIF-8 in our experiments was around 5 μmol/ml, whereas total concentration of amino acids and phosphates in cell media was at least the same (for MEM, 199, alpha-MEM) or even higher (for RPMI, DMEM, IMDM and L15).

New text in SI reads:

“The concentration units of all compounds are 10-5 M.”

New text in manuscript reads:

“It is known that Zn2+ and MIM ligands are released from ZIF-8 in water [53], [54]. In addition, our results showed that the dissolution of ZIF-8 by amino acids is a concentration-dependent process (Figure 4b). The maximum amount of zinc ions which can be released upon complete dissolution of ZIF-8 in our experiments is around 5 μmol/ml, whereas total concentration of amino acids and phosphates in cell media is at least the same (for MEM, 199, alpha-MEM, Table S1) or even higher (for RPMI, DMEM, IMDM and L15, Table S1). Therefore, we assumed that the degradation of nanoparticles is caused by the binding of zinc ions with the amino acids.”

Point 4. Also, a possible impact of Zn(II) complexation by chloride (lines 347-348) should be commented. 

Reply:

The required information has been added

The new text reads:

“We did not considered a possible impact of Zn2+ complexation by chloride in this work because the binding constant β of such a complex is small (Log(β3)<2.5) [61].”

Point 5. I guess authors could be a little more generous in the discussion of amino acid-driven decomposition of ZIF-8. May the highest extent of ZIF-8 destroy by histidine and cysteine be explained in terms of their chemical nature?

Reply:

We added additional discussion of chemical nature of histidine and cysteine complexation with zinc ions

The new text reads:

“High values of histidine and cysteine complexation with zinc ions can be explained by the chemical nature of amino acids in the following way. In case of histidine zinc ions forms a six-membered chelate ring through coordination with two nitrogen atoms of the histidine moiety that could be more stable than analogous one with other amino acids [56]. In contrast, other amino acids form five-membered chelate ring through coordination with nitrogen and oxygen [56], therefore their complexes with zinc ions are less stable compared to histidine. The Sulphur atoms of cysteine have high affinity to zinc ions [57,58]. Thus, cysteine also exhibits high stability of complexes with zinc ions despite five-membered chelate ring.”  

Point 6. A purity and composition of the obtained R@ZIF-8 phase needs to be proven by some conventional characterization, e.g. PXRD/IR/CHN data. 

Reply:

R@ZIF-8 samples were synthesized according to procedure published by us previously [https://doi.org/10.1021/acs.nanolett.9b02730, https://doi.org/10.1021/jacs.8b03584]. PXRD measurements were carried for the radical@ZIF-8 samples in those studies, and phase purity of radical@ZIF-8 samples was approved. The references to these articles were added in the description of the synthesis of R@ZIF-8 samples.

The new text reads

“R@ZIF-8 nanoparticles were prepared according to the procedure described previously [51,59-60]. For the synthesis of ZIF-8 nanoparticles R was dissolved in 2.5 M MIM solution to concentration around 0.0027 M and the mixture was stirred during 30 minutes. Then 0.5 M zinc nitrate solution was slowly added to the above mixture under mechanical agitation for 30 minutes. ZIF-8 nanoparticles were collected by centrifugation and washed with water 3 times. PXRD measurements were carried for the radical@ZIF-8 samples in previous studies [59-60] and phase purity of radical@ZIF-8 samples was approved.

Point 7. Some references should be provided to confirm that 60-70% cell viability represents "only a slight cytotoxicity" (line 288). 

We have reworded this statement to make it more specific. In addition, we changed the conclusion in the same way.

The new text in Line288 reads:

“Additionally, we have shown that 24 h incubation of cells with 10 mM MIM does not decrease cell viability, and there is a moderate cytotoxic effect at 30 mM (60-70% cell viability for 24 h cell incubation, FigureS10).”

The new text in conclusion reads:

“We also found that MIM partially inhibits ZIF-8 dissolution in the DMEM, opti-MEM, and L15 at 10 mM concentration, and it is nontoxic for cells at this amount. Moreover, 30 mM of MIM showed a moderate cytotoxicity for 24 h cell incubation. Thus, MIM can be safely used for ZIF-8 stabilization at these concentrations in short-term cellular studies. Finally, the preliminary addition of 30 mM of MIM to the most widely used DMEM medium significantly reduces cargo release to the same level as in MEM, exhibiting one of the best levels of nanoparticle stability.”

 Some minor corrections: 

Point 7. Author list does not match in the manuscript and ESI. 

Author list has been corrected in ESI

Point 8. Line 152: dilution → dissolution

done

Point 9. Line 166: The introductory phrase is probably missing

We rephrased the first sentence in line 166 to make it clearer.

The new text reads:

“In addition to buffer components, cell media also contain high concentrations of various amino acids (Table S1).”

Point 10. pH-meter should be described in the experimental

Done

The new text reads:

“Measurement of pH was performed using pH electrode InLab Micro (Mettler Toledo) calibrated with buffer standards (pH 4, 7, and 10).”

Reviewer 4 Report

Interesting paper. The stability of MOFs including ZIF series is an important issue in the supramolecular inorganic framework materials area.  The manuscript by Krumkacheva and coworkers presents the stability of ZIF-8 even in the aqueous cell culture media.  They examined the ZIF-8 crystals with EPR in various pH and media to demonstrate its stability, which can allow the use of ZIF-8 in cell experiment sufficiently.  Thus, they finally suggest a guide for selecting MOF materials for cell experiments.  The paper is well organized and written, but some recent papers need to be considered before publication as references as the MOF stability issue are rising one: doi.org/10.1002/bkcs.12268; doi.org/10.1038/s42004-022-00666-8; doi.org/10.1021/jacs.9b02114.